# Identification and validation of ATP6V1G1-regulated phosphorylated proteins in hepatocellular carcinoma

Yi Zhang[1], Liuyi Lu[2], Mingxing Chen[3], Jiaqi Nie[1], Xue Qin[1], Huaping Chen[1]*

**1** Department of Clinical Laboratory, Key Laboratory of Clinical Laboratory Medicine of Guangxi Department of Education Nanning, The First Affiliated Hospital of Guangxi Medical University, Guangxi, China, **2** Department of Clinical Laboratory, Guangxi International Zhuang Medicine Hospital, Nanning, Guangxi, China, **3** Department of Clinical Laboratory, Guangxi Maternal and Child Health Hospital, Nanning, Guangxi, China

* yfy004531@sr.gxmu.edu.cn

**Data Availability Statement:** The data is available through ProteomeXchange at http://proteomecentral.proteomexchange.org/cgi/GetDataset?ID=PXD055721 and iProX at https://

## Abstract

V-ATPase Subunit G1 (ATP6V1G1) is one of the subunits of Vacuolar ATPases. Previous studies have indicated that ATP6V1G1 plays important roles in hepatocellular carcinoma (HCC) and is associated with HCC progression. However, the effect of ATP6V1G1 in HCC requires further elucidation. The aim of the present study was to explore the roles of ATP6V1G1 in HCC and further decipher the detailed mechanism. To identify the phosphorylated proteins regulated by ATP6V1G1 in HCC, phosphoproteomics and LC-MS/MS analysis were performed on HepG2 with overexpression of ATP6V1G1 and empty vector. Western blotting was applied to validate the differentially expressed phosphorylated proteins (DEPPs). As a result, 163 DEPPs were identified by proteomics; two up-regulated phosphorylated proteins (p-RPS6(Ser235)) and (p-SQSTM1(Ser272)) and two down-regulated phosphorylated proteins (p-PDPK1(Ser241)) and (p-EEF2 (Ser57)) were validated. Taken together, this study highlighted the potential impact of ATP6V1G1 on tumor progression, which may be beneficial to liver cancer related basic research.

## Introduction

Hepatocellular carcinoma is the sixth most common cancer and the third leading cause of cancer deaths globally, with approximately 906,000 new cases and 830,000 deaths per year [1]. China is a high-risk area for liver cancer, and the annual new liver cancer patients and liver cancer-related mortality account for 50% of the world's total [2]. Unfortunately, early HCC is difficult to diagnose due to its subtle symptoms and lack of effective biomarkers [3]. In the late stage of tumor, the existing treatment strategies for advanced liver cancer are limited, and the 5-year survival rate of patients with liver cancer is less than 18% [1]. Surgical treatment requires a high level of physical condition for patients, with less than 20% of late stage patients suitable for surgical treatment [4]. Chemoradiotherapy is easy to bring about serious adverse reactions, and the sensitivity of conventional chemoradiotherapy has large individual differences [5,6]. Different from traditional therapies, molecule-targeted therapy designs specific

www.iprox.cn/page/project.html?id=IPX0009689000.

**Funding:** This study was supported by the Youth Science Foundation of Guangxi Medical University (No. GXMUYSF202302), the Joint Project on Regional High-Incidence Diseases Research of Guangxi Natural Science Foundation under Grant (No. 2024GXNSFBA010049, No. 2023GXNSFDA026001), the National Natural Science Foundation of China (No. 81960390). The funders had no role in study design, data collection and analysis, decision to publish, or preparation of the manuscript.

**Competing interests:** The authors have declared that no competing interests exist.

drugs at the cellular and molecular levels according to known carcinogenic sites to achieve highly precise positioning, causing specific death of tumor cells and reducing toxic side effects on normal cells, which is more applicable in clinical practice [7,8]. Therefore, there is an urgent need to understand the molecular mechanism of HCC progression and provide new strategies for targeted therapy of HCC. In recent years, phosphoproteomics studies have shown promise in helping us decipher the molecular drivers of HCC development and leading to new treatment options.

ATP6V1G1 is one of the subunits of V-ATPase, located at 9q32 [9]. V1G1 is part of the peripheral stem of the atpase and is important for proper assembly of the ATP-ase and maintenance of enzyme activity [10]. So far, the specific function of ATP6V1G1 is still unknown. ATP6V1G1 overexpression due to UBQLN2 mutations has been found to cause amyotrophic lateral sclerosis (ALS), frontotemporal dementia (FTD), and other neurodegeneration [11]. Luca et al found that the acid kinetics of Hela cells slowed down regardless of whether ATP6V1G1 was overexpressed or silenced, suggesting impaired lysosomal function [12]. Meanwhile, ATP6V1G1 regulates liver lipid metabolism by maintaining the normal acidification function of lysosomes, suggesting that ATP6V1G1 may play an important role in the development of nonalcoholic fatty liver disease [13]. Up to now, little has been reported about ATP6V1G1 in cancer. Previous studies have shown that high levels of ATP6V1G1 are associated with short overall survival in glioblastoma (GBM) and ATP6V1G1 interacts with other factors to reprogram the surrounding non-tumor microenvironment to a pro-tumor state [14]. In the previous study, ATP6V1G1 was highly expressed in HCC and closely related to tumor cell proliferation, apoptosis, and migration [15]. Recent evidence suggests that V1G1 can perturb protein phosphorylation and phosphorylation-related pathways [16]. However, the molecular mechanism of ATP6V1G1 in HCC remains unclear and needs to be further elucidated.

Protein phosphorylation regulates vital life activities, including cell proliferation, differentiation, apoptosis, and signal transduction [17]. Abnormal changes in protein phosphorylation often accompany the occurence and progrssion of tumors [18]. Given the importance of protein phosphorylation, we employed a phosphoproteomics approach to detect the DEPPs before and after ATP6V1G1 expression alteration in HepG2. This study established the foundation for further investigation of the role of ATP6V1G1 in HCC.

## Materials and methods

### Cell culture

Human hepatoma cell lines HepG2 and Huh7 were obtained from the Cell Bank of Shanghai Academy of Sciences. The cell lines were cultured in DMEM high glucose medium supplemented with 10% fetal bovine serum and 1% cyane-streptomycin. The cultures were maintained at 37°C with 5% CO2.

### Lentiviral transfection

The ATP6V1G1 overexpression lentiviral vector was constructed by GeneCopoeia (USA). HeGp2 and Huh7 cells were transduced using lentiviruses with polybrene (8 μg/ml, GeneChem). After 72 hours of transduction, cells were selected with 5 μg/ml puromycin for a period of 10–20 days.

### RNA extraction and quantitative real-time PCR

Total RNA was extracted from HCC cells using NucleZol reagent (Macherey-nagel, Germany). The cDNA was synthesized using the TaKaRa RNA reverse transcription kit (TaKaRa, Japan) and amplified in an ABI17500 real-time PCR instrument (ABI 7500, USA).

## Western blotting

Total protein was extracted from adherent cells using RIPA lysate (Solarbio, China), containing cocktail protease phosphatase inhibitor (Thermo Scientific, USA) and PMSF pyrolysis. (Solarbio, China). The protein concentration was determined using the BCA assay (Thermo Scientific, USA). The samples were separated by sodium dodecyl sulfate polyacrylamide gel electrophoresis (SDS-PAGE) and transferred to a polyvinylidene difluoride (PVDF) membrane. The membranes were then blocked in TBST solution containing 5% skim milk for 1 hour and with the specific primary antibody overnight at 4˚C. This was followed by incubation with secondary antibodies for 1 hour at room temperature. The primary antibodies used in this study were as follows: anti-ATP6V1G1 (1:500), anti-p-RPS6 (Ser235) (1:1000), anti-p-SQSTM1 (Ser272) (1:1000), anti-p-EEF2 (Ser57) (1:1000), anti-p-PDPK1 (Ser241) (1:1000), and anti-β-actin (1:1000).

## Preparation of phosphorylated peptides

Total proteins were extracted from HepG2 cells overexpressing ATP6V1G1 and empty vector according to the SDS lysate instructions (Biyuntian, China). Protein concentrations were detected using a BCA kit. Each sample was taken 200μg protein reduction and alkylation, and then digested with trypsin overnight at 37˚C. (Sigma, USA). Obtained peptides were labeled with TMT6 labeling reagent. Phosphorylated peptides were enriched using the $TiO_2$ method and separated by chromatograph.

## LC-MS/MS analysis

The phosphorylated peptide segment was isolated using the Easy-nLC1200 liquid chromatograph, the sample was dissolved in liquid phase A ($H_2O$:FA, 99.9:0.1). Then, the sample was loaded onto the pretreatment column Acclaim PepMap100 at a flow rate of 300 nl/min and then separated by the analytical column Acclaim PepMap RSLC. The gradient elution conditions was from 0 to 82 minutes, liquid B (ACN:$H_2O$:FA, 80:19.9:0.1) increased from 5% to 44%; from 82 to 84 minutes, liquid B increased from 44% to 90%; from 84 to 90 minutes, liquid B was maintained at 90%. The specific data settings for mass spectrometry (MS) were listed as follows: in positive ion mode, mass resolution was set to 60,000, and the automatic gain control value was set to 1e6. The MS scan was set to the full-scan range of 300–1600, and tandem mass spectrometry (MS/MS) scans were performed on the 20 highest peaks. All MS/MS pattern acquisitions were accomplished using high-energy collisional dissociation in data-dependent mode with collision energy set to 32. The resolution of MS/MS was set to 30,000, the automatic gain control was set to 2e5, and the dynamic exclusion time was set to 30 secs. The data were imported into MaxQuant software for library searching, and the false-positive rate of peptide identification was controlled to be less than 1%.

## Bioinformatics analysis

The conditions for differential screening were fold change = 1.2-fold and p < 0.05. GO annotation and KEGG pathway enrichment analysis were performed for the differentially expressed phosphorylated proteins.

## Statistical analysis

SPSS24.0 and GraphPad Prism8 software were used for statistical analysis, and the measurement data were presented as mean ± standard deviation. Student's t-test with bilateral significance were used for continuous variables. P < 0.05 was considered statistically significant.

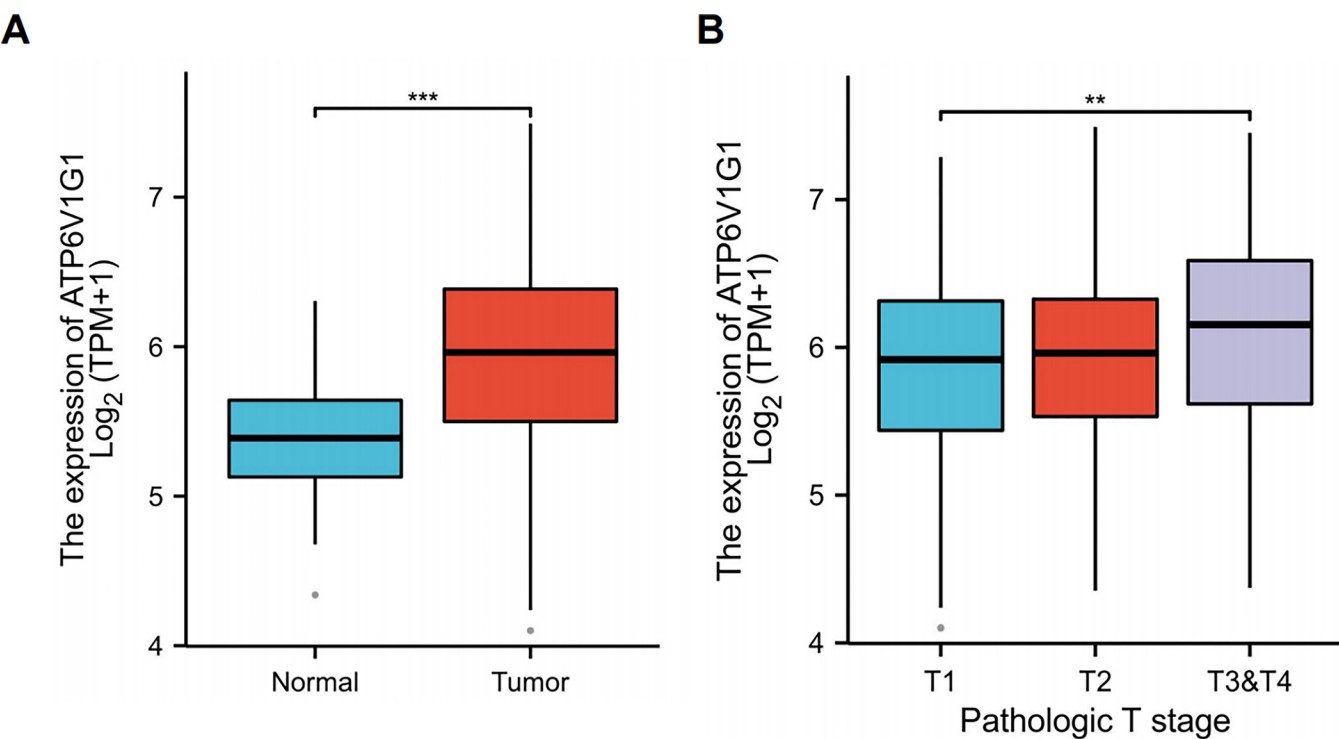

**Fig 1. ATP6V1G1 is highly expressed in HCC compared with normal.** (A) ATP6V1G1 is highly expressed in advanced HCC.(B) ATP6V1G1 HCC cell lines were successfully constructed.

## Results

### Elevated ATP6V1G1 is correlated with the stages of HCC

Public data (https://portal.gdc.cancer.gov) revealed that ATP6V1G1 expression was dramatically increased in HCC compared with normal tissues (Fig 1A). And we observed that, the high expression of ATP6V1G1 is also related to the clinicopathological stage. As shown in the figure, ATP6V1G1 expression was elevated in advanced HCC (Fig 1B). The results indicated that ATP6V1G1 may be associated with the progression of HCC.

The RT-PCR and western blot results showed that the molecular transcription and protein levels of ATP6V1G1 were higher in the lentivirally transfected HepG2 cell overexpression group compared to the empty group (Fig 2A and 2B). This suggests that successful construction of HepG2 and Huh7 HCC cell lines with stable overexpression of ATP6V1G1 was achieved.

### Analysis and validation of differentially expressed phosphorylated sites and corresponding protein expression

The number of detected phosphorylated proteins, peptides, and sites were 1873,4747, and 5897, respectively (S1A and S1B Fig). Compared to the control group, the ATP6V1G1 overexpression group was detected 163 proteins with altered phosphorylation levels and 228 altered phosphorylation sites (Fig 3A), including 88 sites with upregulated phosphorylation levels and 140 sites with downregulated phosphorylation levels (S2 Fig). Table 1 shows the top ten differentially expressed phosphorylated sites and corresponding proteins according to the screening conditions for upregulation and downregulation. Two upregulated phosphorylation sites and

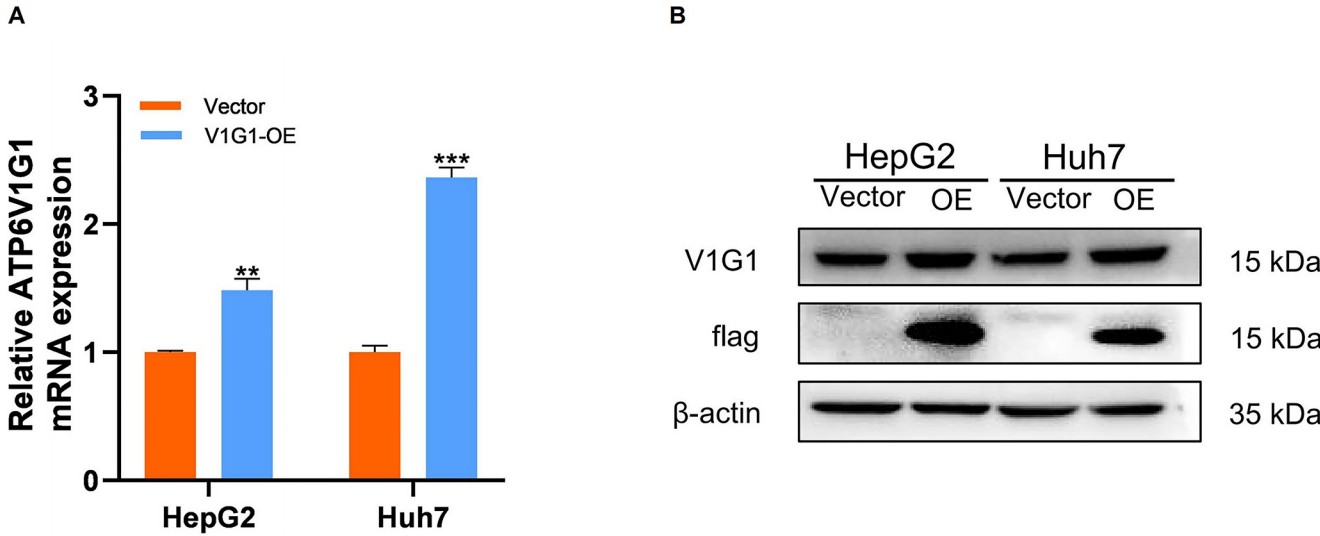

**Fig 2. Verification of overexpression efficiency after lentivirus transfection.** (A.RT-PCR, B.Western blot) ** P < 0.01; *** P < 0.0001.

two downregulated phosphorylation sites and their corresponding proteins were selected for western blot verification. The results showed that p-RPS6 (Ser235) and p-SQSTM1 (Ser272) were upregulated in the ATP6V1G1 overexpression group, while p-PDPK1 (Ser241) and p-EEF2 (Ser57) were downregulated in the ATP6V1G1 overexpression group (Fig 3B). The change trend of these phosphorylated proteins was consistent with the results of phosphorylated proteomics and prompted the reliability of phosphorylated proteomics results.

## Functional analysis of differentially expressed phosphorylated proteins

Analysis of Gene ontology (GO) was performed on the differentially expressed phosphorylated proteins (DEPPs). The top 15 enrichment terms of GO analysis of (DEPPs) were showed in

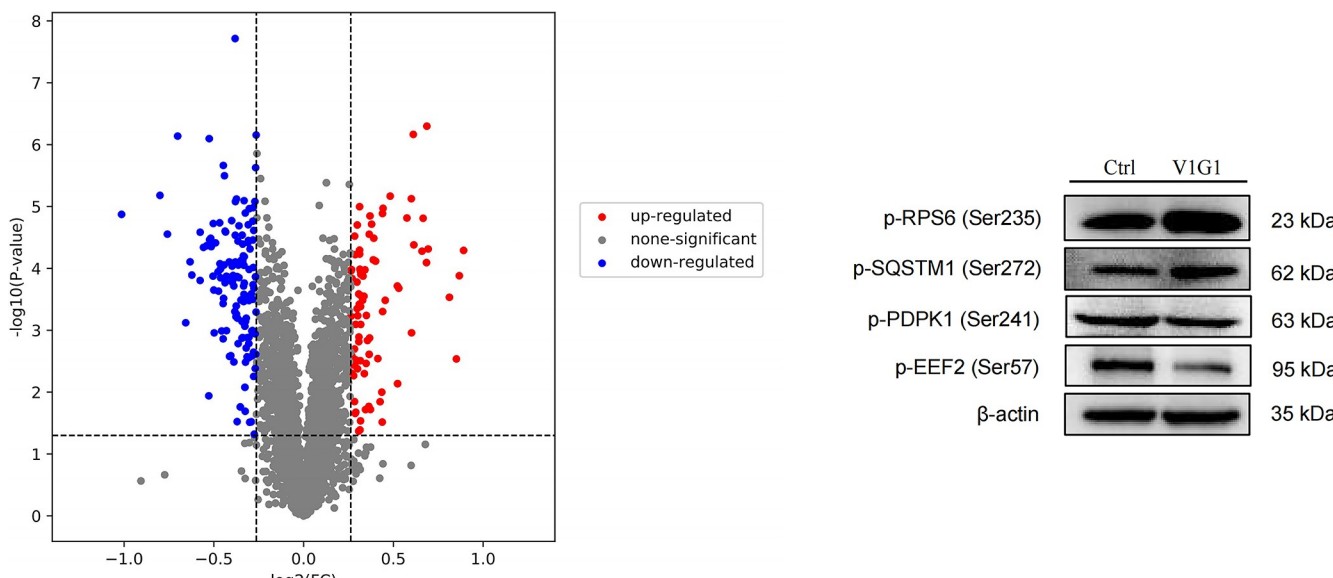

**Fig 3. Volcano plot of differential expression of proteins with altered phosphorylation in the ATP6V1G1 overexpression group.** (A) The differential phosphorylation sites and their corresponding proteins were verified by Western blot (B).

**Table 1. The top ten up-regulated and down-regulated differential phosphorylation sites and their corresponding proteins in phosphorylated proteomics.**

| Proteins | Gene Name | Phospho-site | *P*-value | FC | Expression |
|---|---|---|---|---|---|
| P10412 | H1-4 | Ser27 | 5.03E-07 | 1.609543 | Up-regulated |
| Q01850 | CDR2 | Ser311 | 6.79E-06 | 1.397181 | Up-regulated |
| Q6Y7W6 | GIGYF2 | Ser30 | 7.48E-06 | 1.516089 | Up-regulated |
| P17096 | HMGA1 | Ser36 | 1.01E-05 | 1.241422 | Up-regulated |
| P46087 | NOP2 | Ser732 | 1.07E-05 | 1.358968 | Up-regulated |
| P46013 | MKI67 | Ser859 | 1.30E-05 | 1.356218 | Up-regulated |
| Q9ULW0 | TPX2 | Ser738 | 1.42E-05 | 1.292236 | Up-regulated |
| Q9P2D0 | IBTK | Ser1004 | 1.53E-05 | 1.490388 | Up-regulated |
| P62753 | RPS6 | Ser235 | 1.93E-04 | 1.43635 | Up-regulated |
| Q13501 | SQSTM1 | Ser272 | 2.59E-04 | 1.23725 | Up-regulated |
| Q9UQ35 | SRRM2 | Ser1562 | 1.93E-08 | 0.767627 | Down-regulated |
| Q96ST2 | IWS1 | Ser440 | 6.99E-07 | 0.832691 | Down-regulated |
| Q05209 | PTPN12 | Ser435 | 7.27E-07 | 0.614925 | Down-regulated |
| Q9H1E3 | NUCKS1 | Ser79 | 8.00E-07 | 0.69465 | Down-regulated |
| P08833 | IGFBP1 | Thr193 | 2.17E-06 | 0.733483 | Down-regulated |
| P22314 | UBA1 | Ser4 | 2.36E-06 | 0.830946 | Down-regulated |
| Q8NDI1 | EHBP1 | Ser428 | 3.18E-06 | 0.736698 | Down-regulated |
| O43491 | EPB41L2 | Ser87 | 6.61E-06 | 0.574282 | Down-regulated |
| P13639 | EEF2 | Ser57 | 7.56E-06 | 0.771663 | Down-regulated |
| O15530 | PDPK1 | Ser241 | 2.70E-04 | 0.796138 | Down-regulated |

Fig 4. In biological processes category, DEPPs are mainly involved in negative regulation of transcription from RNA polymerase II promoter, regulation of cellular response to heat, IRE1-mediated unfold protein response and TOR signaling. The cellular component category showed that DEPPs are associated with nucleus, membrane, nucleolus, focal adhesion and cytoplasmic stress granule. The molecular function category suggested that DEPPs are mainly related to cadherin binding involved in cell-cell adhesion, identical protein binding, protein kinase binding and kinase activity.

## Discussion

DNA is transcribed into mRNA and then translated into proteins with specific amino acid sequences to play a role in the body. Most of these proteins often need to be chemically modified to have real activity. This modification is referred to post-translational modification [19]. Among the many types of post-translational modifications, phosphorylation modification is one of the most important covalent modifications in living organisms and is by far the most distributed and investigated modification, which occurs when the phosphoryl group of adenosine triphosphate (ATP) is transferred to the amino acid (serine, threonine, and tyrosine) side chain of a protein under the catalytic action of kinases. The reversible process of protein phosphorylation and dephosphorylation regulates all life activities including cell proliferation, differentiation, apoptosis, development, muscle contraction, neural activity, signal transduction, and tumorigenesis [17].

As an important subunit of V-ATPase, ATP6V1G1 regulates intracellular and extracellular PH balance by affecting V-ATPase activity. Studies have revealed that V1G1 regulates the invasion and metastasis of breast cancer cells [20]. The high expression of ATP6V1G1 promotes the proliferation and migration of glioma stem cells [21]. However, the role of ATP6V1G1 in HCC, as well as the biological functions and related mechanisms are still rarely reported. In

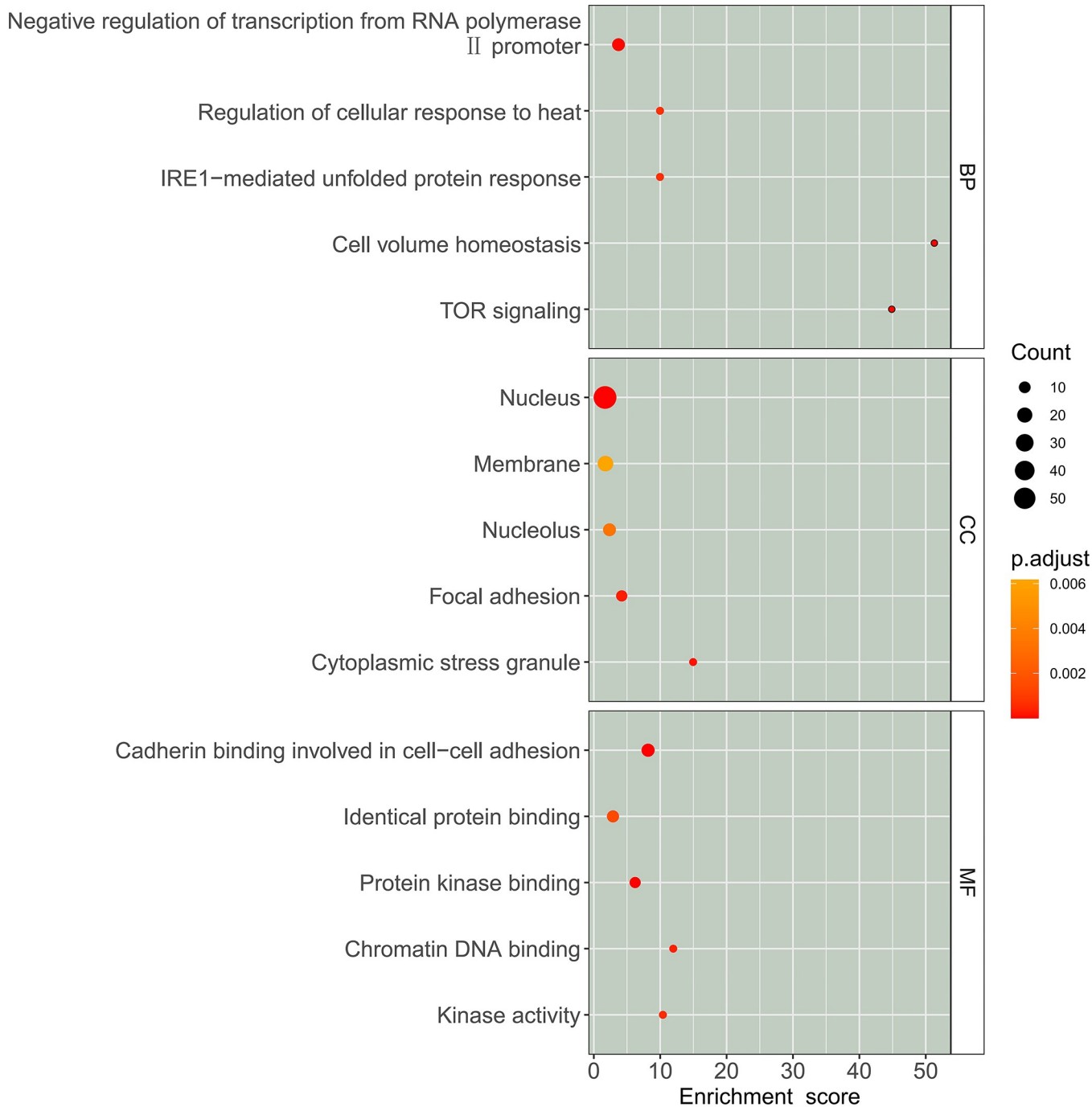

**Fig 4. The 15 top GO terms of differential phosphorylated proteins.** BP, biological process CC, cellular component MF, molecular function.

this study, a comprehensive phosphorylation proteomic study of ATP6V1G1 overexpression of the HepG2 cell line was carried out using TMT-tagging, $TiO_2$ phosphorylation enrichment, and LC-MS/MS proteomic techniques based on TMT-tagging, combined with bioinformatics analysis. A total of 1,873 phosphorylated proteins, 4,747 phosphorylated peptides, and 5,897 phosphorylation sites were identified in this study, which is similar in scale to the results obtained from phosphorylated proteomes of HCC cells reported in the literature [22,23].

The two identified upregulated phosphorylated proteins p-RPS6(Ser235) and p-SQSTM1 (Ser272) and two downregulated phosphorylated proteins p-PDPK1(Ser241) and p-EEF2 (Ser57) were verified by western blotting (Fig 3B). The results were basically consistent with the results of phosphoproteomics, suggesting the reliability of the omics conclusion. The ribosomal protein S6 (RPS6), which is the main substrate for protein kinases in ribosomes, has a subgroup of five c-terminal serine residues that can be phosphorylated by different protein kinases. Growth factors, tumor promoters, and mitogens are inducers of RPS6 phosphorylation. In the signal transduction pathways such as mTOR and PI3K/Akt, mTORC1 further activates RPS6 through phosphorylation of the p70 ribosomal protein S6 kinase to promote protein synthesis for cancer cell growth. SQSTM1 is a key protein in the formation of autophagy, which reflects the activity of autophagy in cells. SQSTM1 was integrated into autophagosomes after binding to the substrate of LC3. When autophagy occurs in cells, the fusion of autophagosomes and lysosomes gradually leads to the formation of autophagolysosomes. Subsequently, lysosomal enzymes degrade SQSTM1 proteins or organelles in vesicles, resulting in a decrease in SQSTM1 levels. Conversely, when autophagy was inhibited, SQSTM1 levels were elevated. In addition, it has been shown that phosphorylation of SQSTM1 is closely associated with cisplatin resistance in tumor cells and the progression of the TAR DNA binding protein (TARDBP) associated neurodegenerative diseases [24,25]. Abnormal cellular metabolism is a key feature of tumorigenesis and development to support its rapid proliferation, migration and escape from apoptosis [26]. The liver is the largest metabolic organ in the body, abnormal expression of metabolism-related enzymes is closely related to primary HCC, and abnormalities in glucose, lipid, and amino acid metabolism may promote tumorigenesis [27–29]. In the early stage of HCC, the biosynthesis of raw materials such as amino acids, nucleotides, and membrane phospholipids increases to meet the needs of cancer cell division and proliferation. Phosphorylation and dephosphorylation serve as key switches that regulate metabolism. Phosphorylation can directly impact metabolic enzymes or regulate their expression, connecting them to a complex signaling pathway network. In addition, it can influence protein function, stability, and interactions with other proteins, which may indirectly impact the expression of additional proteins. By understanding the activation status of these signaling pathways, we can gain insights into the mechanism of primary liver cancer and identify potential therapeutic targets for HCC [30,31]. In this study, both GO and KEGG pathways of differentially expressed phosphorylated proteins were enriched in metabolic pathways (S3 Fig), suggesting that ATP6V1G1 is involved in a variety of metabolic processes, including lipid metabolism, amino acid metabolism, and nucleotide metabolism. Numerous studies have confirmed that the growth of HCC can be significantly inhibited by targeting these metabolic pathways, providing new ideas for the prevention and treatment of HCC [32].

Despite the theoretical advantages of phosphoproteomics in occurrence and progression of HCC, some problems still need to be solved in practical application. For example the complex composition of tumor tissue, and the phosphorylation of key proteins in HCC cells cultured in vitro may change based on the changes in the cell culture environment. Therefore, in vivo experiments are needed to further validate the cellular experiments. Another shortcoming is that while we have identified the proteins that become phosphorylated in response to ATP6V1G1 overexpression in hepatocellular carcinoma (HCC) cell lines, we have not delved into the direct effects of these phosphorylation alterations on the overall protein expression profile. Hence, it is imperative that future studies delve deeper into this area.

In this research, proteomics is primarily utilized to identify potential new players in the progression of hepatocellular carcinoma (HCC) through differentially expressed protein phosphorylation (DEPPs). Our results demonstrate that ATP6V1G1 may regulate multiple cellular processes via phosphorylation and could serve as an important starting point for studying

HCC-specific phosphorylation events.We have demonstrated that ATP6V1G1 regulates the cellular activities of HCC cells via modulating protein phosphorylation and phosphorylation-related pathways. The differentially expressed proteins (DEPPs) identified in this study were primarily enriched in metabolic pathways, as indicated by Gene Ontology (GO) and Kyoto Encyclopedia of Genes and Genomes (KEGG) analysis, suggesting the involvement of ATP6V1G1 in multiple metabolic processes. Therefore, targeting these metabolic pathways to suppress HCC cell growth represents a novel therapeutic approach for HCC treatment.

## Supporting information

**S1 Fig. The number of phosphorylated proteins, phosphorylated peptides, and phosphorylation site was identified.** A. Number of proteins, peptides and sites with altered phosphorylation. B. The proportion of threonine, tyrosine and serine amino acid sites changed by phosphorylation.
(TIF)

**S2 Fig. Differential phosphorylation proteins in phosphorylated proteomics.**
(TIF)

**S3 Fig. KEGG signaling pathway enrichment annalysis results.**
(TIF)

**S1 File. Uncropped western blots.**
(ZIP)

## Acknowledgments

I wish to express my deepest appreciation to Shanghai Luming Biological Technology Co., Ltdfor their exceptional technical support in the field of phosphoproteomics. Their advanced technologies and expert guidance were pivotal in enabling our research on protein phosphorylation. The high-quality services and resources provided by Shanghai Luming Biological Technology Co., Ltd. have significantly contributed to the progress and success of this study. Thank you for your invaluable contributions.

## Author Contributions

**Formal analysis:** Liuyi Lu, Mingxing Chen, Jiaqi Nie, Xue Qin.

**Project administration:** Huaping Chen.

**Writing – original draft:** Yi Zhang.

**Writing – review & editing:** Yi Zhang, Liuyi Lu, Mingxing Chen, Jiaqi Nie, Xue Qin, Huaping Chen.

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
