## [Editor Report · Decision Letter 0]

21 Jun 2024

PONE-D-24-24217Identification and Validation of ATP6V1G1-regulated Phosphorylated Proteins in Hepatocellular CarcinomaPLOS ONE

Dear Dr. Chen,

Thank you for submitting your manuscript to PLOS ONE. After careful consideration, we feel that it has merit but does not fully meet PLOS ONE’s publication criteria as it currently stands. Therefore, we invite you to submit a revised version of the manuscript that addresses the points raised during the review process.

We look forward to receiving your revised manuscript.

Kind regards,

John Wing Shing Ho

Academic Editor

PLOS ONE

Journal Requirements:

"This study was supported by the Youth Science Foundation of Guangxi Medical University (GXMUYSF202302), the Natural Science Foundation of Guangxi (2023JJB140556),  the National Natural Science Foundation of China (81960390), and the Joint Project on Regional High-Incidence Diseases Research of Guangxi Natural Science Foundation under Grant (2024GXNSFBA010049)."

6. Please include a separate caption for each figure in your manuscript.

7. We note that Figure 2 in your submission contain copyrighted images. All PLOS content is published under the Creative Commons Attribution License (CC BY 4.0), which means that the manuscript, images, and Supporting Information files will be freely available online, and any third party is permitted to access, download, copy, distribute, and use these materials in any way, even commercially, with proper attribution. For more information, see our copyright guidelines: http://journals.plos.org/plosone/s/licenses-and-copyright.

Additional Editor Comments:

The manuscript reported the effect of ATP6V1G1 in HCC and protein expression associated with of ATP6V1G1 activity. The phosphorylated proteins regulated by ATP6V1G1 in HCC were identified by phosphoproteomics and LC-MS/MS analysis. How did the phosphorylated proteins (DEPPs) contribute to the protein expression? How was the impact of ATP6V1G1 on tumor progression deduced based on the present studies? More alternate studies are needed to provide supportive data prior to drawing a convincing conclusion. How do the present results contribute to the mechanistic action of cancer development?

---

## [Author Response · Author response to Decision Letter 0]

25 Jul 2024

Dear Editor and Reviewers,

We are very grateful for the questions you have raised. After thorough and careful consideration, we would like to offer the following responses:

Comment 1: How did the phosphorylated proteins (DEPPs) contribute to the protein expression?

Response: In this study, we concentrate on examining the changes in protein phosphorylation levels associated with the overexpression of ATP6V1G1. As we all know, protein phosphorylation, a pivotal post-translational modification, plays a crucial role in regulating various cellular processes, including protein folding, subcellular localization, enzyme activity, and substrate interaction. This modification can influence protein function, stability, and interactions with other proteins, which may indirectly impact the expression of additional proteins. However, a limitation of our research is that while we have identified the proteins that become phosphorylated in response to ATP6V1G1 overexpression in hepatocellular carcinoma (HCC) cell lines, we have not delved into the direct effects of these phosphorylation alterations on the overall protein expression profile.

Comment 2: How was the impact of ATP6V1G1 on tumor progression deduced based on the present studies? More alternate studies are needed to provide supportive data prior to drawing a convincing conclusion.

Response: Available public data indicate that ATP6V1G1 is upregulated in hepatocellular carcinoma (HCC) cells and correlates with advanced disease stages. Additional research has shown that ATP6V1G1 expression in HCC cells is linked to apoptosis, potentially offering a protective mechanism against cell death, and a reduction in its expression is associated with diminished cell migration. These findings suggest that ATP6V1G1 is a critical factor in regulating cell growth, migration, and invasion, as previously noted in our manuscript. In our investigation, we observed alterations in the phosphorylation levels of proteins that are key players in essential cellular processes such as transcriptional regulation, stress response, and signaling cascades. Furthermore, the Gene Ontology (GO) and Kyoto Encyclopedia of Genes and Genomes (KEGG) pathway analyses of these differentially phosphorylated proteins revealed significant enrichment in metabolic pathways, indicating a role for ATP6V1G1 in diverse metabolic functions. Consequently, we propose that ATP6V1G1 may modulate metabolic pathways in cancer cells through the manipulation of protein phosphorylation events, thereby influencing tumor progression.

Comment 3: How do the present results contribute to the mechanistic action of cancer development?

Response: The primary application of proteomics in this study has led to the identification of novel factors potentially involved in the progression of hepatocellular carcinoma (HCC) – the differentially expressed protein phosphorylations (DEPPs). Our findings suggest that ATP6V1G1 may modulate a variety of cellular processes through phosphorylation events, offering a critical starting point for understanding hepatocellular carcinoma-specific phosphorylation alterations. Nonetheless, as you have correctly noted, further research is essential to provide a more holistic understanding of the relationship between ATP6V1G1 and the underlying mechanisms of tumor development.

Best regards

Huaping Chen

Department of Clinical Laboratory, the First Affiliated Hospital of Guangxi Medical University

yfy004531@sr.gxmu.edu.cn.

---

## [Editor Report · Decision Letter 1]

23 Aug 2024

Identification and Validation of ATP6V1G1-regulated Phosphorylated Proteins in Hepatocellular Carcinoma

PONE-D-24-24217R1

Dear Dr. Chen,

We’re pleased to inform you that your manuscript has been judged scientifically suitable for publication and will be formally accepted for publication once it meets all outstanding technical requirements.

Kind regards,

John Wing Shing Ho

Academic Editor

PLOS ONE

Additional Editor Comments (optional):

The manuscript has been sufficiently improved and is acceptable.
---

## [Editor Report · Acceptance letter]

19 Nov 2024

PONE-D-24-24217R1 

PLOS ONE

Dear Dr. Chen, 

I'm pleased to inform you that your manuscript has been deemed suitable for publication in PLOS ONE. Congratulations! Your manuscript is now being handed over to our production team.

Kind regards, 

on behalf of

Professor John Wing Shing Ho 

Academic Editor

PLOS ONE